# Multiscale Wettability of Microtextured Irregular Surfaces

**DOI:** 10.3390/ma17235716

**Published:** 2024-11-22

**Authors:** Katarzyna Peta

**Affiliations:** Institute of Mechanical Technology, Poznan University of Technology, 60-965 Poznan, Poland; katarzyna.peta@put.poznan.pl

**Keywords:** contact angle, wettability, surface topography, multiscale analysis, electrical discharge machining

## Abstract

Surface microgeometry created by the energy of electric discharges is related to surface wetting behavior. These relationships change depending on the scale of observation. In this work, contact angles correlated with the surface complexity of AA 6060 after electro-discharge machining were analyzed at different observation scales. This research focuses on the methodology of selecting the best scales for observing wetting phenomena on irregular surfaces, as well as indicating the topographic characterization parameters of the surface in relation to the scales. Additionally, the geometric features of the surface that determine the contact angle were identified. In this study, the surfaces of an aluminum alloy are rendered using focus variation 3D microscopy and described by standardized ISO, area-scale, and length-scale parameters. The research also confirms that it is possible to design surface wettability, including its hydrophilicity and hydrophobicity, using electrical discharge machining parameters. The static and dynamic behavior of liquids on surfaces relevant to contact mechanics was also determined.

## 1. Introduction

Surface microgeometry, next to the chemical properties of the material, is one of the most important determinants of surface functional properties such as wetting and lubrication [1]. The behavior of liquids on surfaces can be modeled via specific texturing of the surface microgeometry [2]. Anisotropic textures, including grooved ones, provide anisotropic surface wetting [3], while isotropic textures are characterized by the same wettability in all directions of the surface [4]. Each surface texturing technique can create unique surface features that distinguish these techniques [5]. The regularity of the distribution of geometric features is also important in wetting the surface, significantly influencing the modeling of the behavior of the liquid droplets on the surface [6]. The wettability of surfaces with irregular geometric features is not constant along the entire surface.

Wetting studies are important in systems with solid–liquid contact, including self-cleaning surfaces [7], heat exchangers [8], human joints [9], gear boxes [10], boats [11], and more. Wetting research on engineered surfaces draws inspiration from the behavior of natural surfaces found in nature. The main examples are superhydrophobic lotus leaves and hydrophilic plant-derived fibers. Based on patterns from nature, modern engineering surfaces are textured to resemble naturally occurring surfaces and, in many cases, also coated with bio-inspired materials [12]. The main reason for the hydrophobicity of many natural leaf surfaces is the presence of microbumps [13]. Such observations suggest that the creation of surface textures with microroughness can lead to an increase in the contact angle of engineering surfaces.

Surface wetting is mainly described by the contact angle measured between the tangent to the liquid droplet and the solid surface, using the phenomena of surface tensions between these phases [14]. Young’s basic model assumes that a drop is placed on a perfectly smooth surface. However, this is a simplified model that is not fully adequate to the real, engineering surfaces obtained in manufacturing processes. The Wenzel and Cassie–Baxter models consider surface roughness a factor influencing contact angles. In Wenzel’s model, a drop of liquid fills the topographical valleys of the surface. The coefficient of the averaged surface roughness is taken into account in the calculation of the contact angle [15]. In the Cassie–Baxter model, a drop of liquid is deposited at the peaks of the surface topography without filling its valleys. In this theory, regions of the surface with different roughness can be considered [16]. Another approach that takes into account surface microgeometry in wetting analysis is the surface composition concept that includes the density asymmetry of the liquid–gas phases, as well as the van der Waals interactions of the liquid–solid phases. One of the assumptions of this concept is that the occurrence of multiple energy minima in the total surface free energy leads to contact angle hysteresis. This allows for the development of classical Young’s law and consideration of the density ratio, van der Waals interaction, and intermolecular potential [17]. In practice, metastable contact angle states can be observed. In this way, liquid droplets can be observed to be temporarily consistent with the Cassie–Baxter model, sessile on the surface peaks, before transitioning to an equilibrium state consistent with the Wenzel model, which fills the surface valleys [18]. Wang et al. concluded that there could be other metastable states beyond the transition from the Cassie–Baxter model to the Wenzel model caused by energy barriers depending on the droplet volume, shape, dimensions, and distribution of surface heterogeneity [19]. One way to achieve the global equilibrium of drops that tend to occur in metastable states is to subject the surface to vibrations that allow the drop to achieve a stable shape [20].

Surface Free Energy (SFE) indicates the surface tension of the solid phase, which is one of the determinants of surface wettability [21]. Certain surface modification methods not only change the surface microgeometry but also affect the surface free energy. Some of the methods for changing the SFE are electrical discharge machining, plasma activation, chemical action of acids, or the biological deposition of peptides and proteins [22]. High surface free energy is associated with good wettability, which is particularly important in coating, painting, and bonding processes [23]. In some applications, low surface free energies are desired to reduce wettability. This is particularly important for surfaces exposed to moisture and corrosion [24].

Texturing is one of the techniques for changing the wettability of the surface. Geometric features of the surface on the nanoscale can already affect the functional features of the surface on the macro scale. This means that even a nano-roughness can determine the wettability of the surface observed on the macro scale [25]. Processes suitable for modeling surface wetting are electrical discharge machining, sandblasting, laser texturing, and other processes, all of which enable the creation of geometric features of the surface [26]. The existence of correlations between texturing parameters and surface topography makes it possible to model surface wettability [27].

Surface wettability studies are primarily based on topographic characterization of the wetted surface. The surface topography characterization parameters described in ISO 25178-2 are the standard for describing engineering surfaces [28]. However, fluctuations in nanometric terms can occur between the surface topographic characterization parameters depending on the method and mode of topographic measurement. There is a premise that different surface topography measurements can yield differences in correlations with the contact angle [29]. It is important to identify the sizes and shapes of geometric features that can significantly change the surface wettability [30,31]. Surface roughness can be seen as a landscape of peaks and valleys, the complexity of which can be considered dependent on the fractal dimension [32]. These relationships are the basis of multiscale analyses [33,34], indicating the best scales of observation of surface microgeometry and then their functional correlations, including with wetting. The aim of multiscale analyses is to determine the boundary scale of observation from which the fractal dimension of the surfaces oscillates around the Euclidean, which indicates the observation of the surface from too far away. The next goal is to find scales that reliably describe the surface and can indicate functional relationships with wettability [35].

The multiscale approach to surface wetting is an evolving area of study. Wang et al. and Armstrong et al. presented the wetting of porous media used in geoscience applications. The authors indicated directions for further research, suggesting measuring the contact angles of porous surfaces depending on the pore scale [36,37]. Gim et al. described a method for predicting surface wetting at the atomic scale, which can be translated into macroscale wetting phenomena [38]. Jain et al. proposed a fractal model of wettability on randomly textured surfaces with multiscale geometric surface features [39]. Davis et al. showed no relationship between the fractal dimension and the contact angle on laser-textured surfaces [40]. Hatte et al. presented multiscale hydrophobic surfaces using fractals to evaluate liquid flow and heat transfer. Kant et al. described multiscale texturing of fractal surfaces for photovoltaic applications and the impact of texturing on wettability [41]. Recent scientific works combine the consideration of surfaces as fractal and correlating them with the functional features of the surface, including wettability. Relationships between the multiscale nature of surface geometric features and the behavior of liquid droplets on surfaces are also considered important.

The novelty of this research includes the identification of the best scales for observing the wetting phenomena of irregular textured surfaces. This approach includes the indication of scale-sensitive dependencies between the fractal complexity of irregular surfaces and contact angle. It was also considered important to indicate the scales of surface wetting observations in linear and areal terms. The aim of this research includes the indication of determination coefficients R^2^ between the conventional (ISO 25178-2) and multiscale parameters (ASME B46.1 [34]) of surface topographic characterization and contact angles. The coefficients of determination between the discharge energy and the parameters of the surface topographic characterization are also calculated. The scope of the determination coefficient analyses allows us to indicate the interactions of three dependent components: texturing, surface, and wettability. This work also includes an analysis of factors that have a significant impact on surface wettability interactions, such as droplet spreading time on the surface and droplet volume.

## 2. Materials and Methods

Experimental studies were performed for AA 6060 T4, textured by electro-discharge machining using Agie Charmilles Form 20, GF Machining Solutions (Biel/Bienne, Switzerland). EDM was performed on 9 rectangular parts previously milled with dimensions of 40 × 40 × 5 mm. Distilled water was used as the dielectric liquid in the texturing process. Electric discharges were conducted between the copper electrode and the textured surface of the aluminum alloy. During texturing, a stable position in the x and y axes of the electrode and the workpiece material was ensured. EDM parameters for surface texturing are shown in Table 1. The schematic view of the electro-discharge machining is shown in Figure 1.

The selection of EDM parameters was based on the internal machine control system. Selection of the expected average surface roughness resulted in the automatic selection of machining parameters. The aim of surface texturing was to obtain a wide range of average surface roughness from approximately 1 µm to 30 µm. Discharge energy E is one of the most important characteristics of electrical discharge machining that affects the size of the created craters and surface roughness. Discharge energy is calculated based on EDM parameters: current, voltage, and pulse on time [42].

Both surface topography and wettability measurements were preceded by cleaning the parts each time, according to the following scheme: 0.5 min removal of contaminants with compressed air, 5 min immersion of the parts in isopropyl alcohol in an ultrasonic cleaner, and 1 min immersion in acetone in an ultrasonic cleaner. Measurements were taken right after surface cleaning. The ambient temperature during the studies was 21 °C.

Surface topographies were measured using a Bruker Alicona 3D optical microscope (Bruker, Billerica, MA, USA) in the focus variation mode. The surfaces were measured in 3 random areas in the middle part. Topographic measurement parameters were set the same for each surface: magnification 20×, lateral dimensions 2800 × 2800 µm, lateral sampling interval 0.49 µm, lateral resolution 2.9 µm, and vertical resolution <50 nm. The surface was measured as a point cloud (x, y, z) and then processed in the MountainsMap 9 software from Digital Surf. Surface processing included leveling the surface using the least squares method, thresholding and removing outliers, and filling non-measured points. Then, the parameters of topographic surface characterization from height and hybrid groups were determined in accordance with the ISO 25178 standard [28].

Surface wettability was measured using an OCA 15 Pro goniometer from DataPhysics (Filderstadt, Germany). The measuring liquid was distilled water dosed in volumes of 1, 2, and 3 µL at a rate of 1 µL/s. For each surface, 7 contact angle measurements were performed and then averaged. The captured images of the sessile droplet on the surface were processed in DataPhysics SCA 20 software. The contours of the liquid droplet were outlined using the Young–Laplace method. Wettability measurements included determining the contact angle versus time as the drop settled on the surface, as well as calculating the surface free energy using the Owens–Wendt method. The liquids used to determine the SFE were distilled water as a polar liquid and diiodomethane as a dispersive liquid.

Multiscale analyses, at both the area scale and length scale, were performed to indicate the relationship between the coefficient of determination R^2^ regarding surface fractal complexity and the features determining wettability, relative to the scale. The concept of scale in multiscale analyses is related to the patchwork method, which consists of filling the surface with equal, triangular tiles (scale). Multiscale analyses distinguished four scale-sensitive parameters [34]:Relative area (Srel) is equal to the calculated area (CA)/nominal area (NA).Relative length (RL) is equal to the calculated length (CL)/nominal length (NL).Area-scale fractal complexity (Asfc) is equal to 1000 times the slopes appearing in log-log relative area plots.Length-scale fractal complexity (Lsfc) is equal to 1000 times the slopes appearing in log-log relative length plots.

Figure 2 presents a diagram of the research carried out within these studies.

## 3. Results and Discussion

### 3.1. Conventional and Multiscale Surface Characterizations

Electro-discharge machining allowed us to obtain irregular surfaces (S1–S9) with various crater sizes depending on EDM parameters, mainly discharge energy (Figure 3). Three-dimensional images of the surface, along with the basic topographic surface characterization parameters, are shown in Figure 4.

Surface microtexturing is an effective way to model wettability and lubrication. Geometric features of the surface on a micro scale affect functional phenomena of the surface, even those visible on a macro scale, such as wetting. One of the machining processes that allows for obtaining irregular surfaces but effectively changing wettability is electro-discharge machining. The selection of EDM parameters and calculation of discharge energy based on them allows for the creation of craters of various sizes on the surface. Craters can perform various functions, including being lubricant reservoirs useful in friction conditions of two cooperating mechanical parts or hindering the flow of liquids on the surface, which is especially important in applications requiring hydrophobic surfaces. Increasing the discharge energy creates larger craters, which can be described by parameters of the surface topography characterization in accordance with the ISO 25178-2 standard. The parameters of surface topography characterization that best correlate with discharge energy are the arithmetical mean roughness Sa, root mean square height Sq, maximum pit height Sv, and maximum height Sz. The coefficient of determination R2 between these parameters and the discharge energy is greater than 0.8. However, a weak correlation is observed for skewness Ssk and kurtosis Sku (R2 is less than 0.15). The discharge energy can therefore more easily shape the mean surface roughness, peak height, and valley depth than the symmetry of the surface height (Ssk) or the peaks and valleys that deviate from the mean plane (Sku). The distribution of craters created by the energy of discharges is irregular but without valleys and peaks that stand out excessively from the average height plane.

Table 2 provides the coefficients of determination R^2^ between the discharge energy per pulse E and the height and hybrid parameters of the surface. 

Figure 5 presents four scale-sensitive surface parameters: relative area, relative length, area-scale fractal complexity, and length-scale complexity. Multiscale parameters [34], similarly to conventional topographic surface characterization parameters [28], allow for distinguishing EDM surfaces treated with different electrical discharge energies. The relationships of multiscale surface parameters with respect to scale are presented in log-log plots. Relative area defines the relationship of the surface represented by the triangle mesh to the nominally defined area. Area-scale fractal complexity can be compared to the degree of detail of microgeometric features of the surface. Length-scale parameters have an analogous meaning, but instead of 3D areas of the surface, they describe 2D profiles.

EDM-machined surfaces are distinguishable not only by most conventional surface characterization parameters but also by the multiscale parameters relative area, relative length, area-scale fractal complexity, and length-scale complexity. These multiscale parameters can be used to mathematically discriminate surfaces at certain scales of observation. The concept of scale in surface metrology can be interpreted as the ratio of lengths on rendered surface images to actual lengths on actual surfaces. However, there is also another meaning for the scale used in scale-sensitive fractal analyses, where scale refers to a segment of wavelengths or spatial frequencies [33]. Each surface can be built with triangular tiles with a fixed area (certain scale). Depending on the technique used to simulate the surface with different sizes of triangles, different values of surface wetting can be obtained. Interestingly, when analyzing irregular surfaces, these are neither the smallest triangles covering the surface nor the largest ones. There is a certain scale of observation and triangle size that best allows for the correlation of surface complexity parameters with the contact angle. It is clear that not all scales distinguish the fine microgeometric features of the surface created by the discharge energy, which later affects wettability. Similarly, in the case of too-fine scales, the visibility of too-detailed microgeometric features of the surface can interfere with the interpretation of the effect of EDM on the surface and, consequently, also of the wetting phenomena. Thus, the multiscale parameters, both the area scale and length scale, allow us to define the range of scales at which surfaces of different surface microgeometry are distinguishable. Relative area and relative length are largest at small scales; then, EDMed surfaces are best distinguishable. As the scale increases, surface distinguishability deteriorates. Fractal complexity only allows us to discriminate surfaces in a certain range of scales; for the area scale, it is the range of 100–10,000 µm^2^, and for the length scale, 10–100 µm.

### 3.2. Surface Wettability

Scale-sensitive surface wetting analyses were preceded by an assessment of liquid droplet spreading in relation to time (Figure 6) and a comparison of three recommended droplet volumes from the literature on contact angle (Figure 7). Both studies aimed to verify two important factors (the spreading and volume of the droplet) to determine surface wetting. Both the spreading and volume of the droplet are characterized by a clearly visible trend. The time taken into account in the analysis of the spreading contact angle covers the range from the moment of stabilization of the droplet deposited on the surface (0 s) to 700 s. This range allows for the observation of the behavior of the liquid droplet from deposition to complete spreading on the surface. Contact angles are measured after stabilization of the droplet deposited on the surface. Based on the experiment, it is determined that after 2 s from the moment of contact of the droplet with the surface, it reaches a stable shape.

In the first 600 s, after the drops are deposited on the surfaces, a decreasing trend in the contact angle can be observed. In this time range, the same relationship between contact angles and surface complexity is clearly visible, both right after droplet deposition on the surface and during droplet spreading on the surface. From about 600 s, the droplets spread, and the contact angles for all surfaces almost equalize. The volumes of droplets 1, 2, and 3 µL do not significantly affect the contact angle. The standard deviation between the contact angles for the three analyzed droplet volumes does not exceed 1°.

The phenomenon of spreading is related to contact angle measurements. It is important to maintain the same contact angle measurement time from the moment the drop settles on the surface for all measurements. With time, the sessile drop spreads, which reduces the contact angle. It is also possible for droplets to partially evaporate, which also contributes to maintaining similar atmospheric conditions, temperature, and humidity in the laboratory room. It is worth preceding static contact angle measurements with droplet spreading experiments over time to verify the time frame of measurements that will allow us to obtain reliable results. Approximately 600 s after the droplets have settled on the surface, their contact angle gradually decreases until they completely spread. All contact angle measurements were taken 2 s after the droplet deposition on the surface. This allowed the droplet to stabilize on the surface and, at the same time, not to start the spreading process.

One of the factors that can affect the contact angle results is the drop volume. A drop that is too large can spread prematurely over the surface due to the effects of gravity or capillary forces. On the other hand, a drop that is too small may not completely fill the surface craters and may not correlate with the surface microgeometry. The selection of the drop volume was based on the literature, and in addition, the effect of three drop volumes, 1, 2, and 3 µL, on contact angles was compared. No significant difference was observed between droplet volumes and contact angles (standard deviations less than 1°).

Wettability depends not only on the surface microgeometry but also on the surface energy. Therefore, SFE analyses are often used to supplement the inferences about surface wettability. Figure 8 shows the surface free energy, divided into polar and dispersive parts, for the analyzed EDMed surfaces, each machined with increasing discharge energy per single pulse. Calculations based on dispersion and polar components allowed us to determine the surface free energy based on the Owens, Wendt, Rabel, and Kaelble (OWRK) method. The discharge energy can affect the energy state of the surface. Surface energy tends to decrease with larger surface craters. The SFE ranges from about 30 mJ/m^2^ for surfaces with larger crater sizes and about 60 mJ/m^2^ for the smoothest EDMed surface.

Preliminary studies of wetting (droplet spreading over time, droplet volume vs. contact angle) and surface free energy indicate a good dependence of contact angles on EDMed surfaces. Figure 9 shows the shapes and contact angles of 1 µL droplets on microtextured surfaces (real views of droplets outlined using the Young–Laplace method). These results provide a basis for calculating multiscale dependencies of wetting and surface complexity.

The microtexturing of surfaces allows for changing the wettability, and furthermore, hydrophilic surfaces can be modified to hydrophobic ones. Therefore, knowledge of the relationships between EDM parameters, surface topographic characterization parameters, and wetting parameters can provide a basis for modeling these dependencies. For a surface with an average roughness of Sa = 1.6 µm, the contact angle is 68°, increasing to 137° at Sa = 30 µm. There is a noticeable trend towards a hydrophilic-hydrophobic transition. The average roughness Sa alone does not is the only parameter determining the contact angle. Based on the determination coefficients between contact angles and surface topographic characterization parameters, the parameters that have the greatest influence on wetting are Sa, Sq, Sp, Sz, Sdq, and Sdr (R^2^ > 0.8). Therefore, a wide group of surface topographic parameters that affect the contact angle and whose acquisition during texturing should be taken into account in order to model surface wettability. Table 3 presents the coefficients of determination R^2^ between the contact angles (CAs) and the conventional surface characterization parameters. 

EDM parameters, mainly discharge energy calculated based on them, affect surface microgeometry. One of the factors influencing wettability is surface microgeometry. Therefore, different EDM parameters modify the microgeometry of the surface and affect the different behavior of liquid droplets on these surfaces. Moreover, the calculated coefficients of determination between interactions and discharge energy in the texturing process, as well as conventional and multiscale parameters of surface characterization and wetting, indicate the existence of these interdependencies.

### 3.3. Multiscale Relationships Between Surface Characterization and Wettability

Indication of the best scales of observation of wetting of microtextured surfaces involves the calculation of the coefficient of determination between contact angles and multiscale surface parameters: relative area, relative length, area-scale fractal complexity, and length-scale fractal complexity. The contact angle vs. area-scale relationships are shown in Figure 10a, and the length-scale relationships in Figure 10b.

Identifying the best scales of observation of wetting of microtextured surfaces allows us to determine the trend of the relationship between the contact angle and the surface fractal complexity according to the strong coefficient of determination R^2^ > 0.96 (Figure 11). The best observation scale for area-scale complexity is 1024 µm^2,^ and for length-scale complexity, it is 32 µm.

Both area-scale and length-scale analyses are effective in determining the best scales for observing wetting phenomena on microtextured surfaces. The coefficient of determination R^2^ > 0.9 between contact angles and relative area occurs for all fine scales up to 100 µm^2^, while the best scale indicated by R^2^ > 0.97 between contact angles and area-scale fractal complexities is 1024 µm^2^. Similar relationship conclusions can be drawn from length-scale analyses. For small scales up to 10 µm, the coefficient of determination between contact angles and relative length is greater than 0.9, while R^2^ between contact angles and length-scale fractal complexity is the largest at the scale of 32 µm and equals 0.97. For the best scales of observation, determined by the largest coefficient of determination, it is possible to draw the most reliable relationship between fractal complexity and contact angle. As fractal complexity increases, so does the contact angle in a linear trend.

Figure 12 presents surface images at various selected observation scales in the form of a triangle mesh and a 3D topographic map of the surface. The comparison includes the best area scale (R^2^ = 0.967) for observing the behavior of liquid drops on a surface of 1024 µm^2^, comparing two other scales (smaller and larger) with worse determination coefficients, namely the 130 µm^2^ scale (R^2^ = 0.62) and the 13,000 µm^2^ scale (R^2^ = 0.73). The presentation of surfaces at different scales aims to visually identify the surface characteristics important from the point of view of wetting. This allows us to indicate visually what level of surface detail leads to the best observation of the behavior of liquid drops on surfaces. The comparative table presents three sample surfaces (S1, S6, and S9), which are representative of the entire range of surface characterization parameters analyzed.

The mathematical characterization of irregular surfaces at different scales is an important tool for analyzing microtextured surfaces and their functional properties, such as wetting and lubrication. Most surfaces analyzed for wetting have irregular microgeometric features, which, despite their complexity, can be characterized by self-similar fractals with respect to scales. The key to this approach is the smooth-to-rough transition (SRC). This means moving from regular Euclidean geometries describing smooth surfaces towards fractals capable of describing rough surfaces. Therefore, the description of microtextured surfaces with characteristic EDM craters can be made possible and effective by using fractal geometries. Increasing the surface irregularity results in larger relative areas and relative lengths at finer scales. Surface complexities are more visible in specific scale ranges, the finding of which is one of the goals of multiscale analyses.

The coefficients of determination between multiscale parameters and contact angle show mathematical relationships. In order to additionally visualize surfaces at certain scales, 3D images of these surfaces are presented together with corresponding triangle meshes covering the surfaces. The images include surfaces S1, S6, and S10 rendered for the best area scale 1024 µm^2^ (R^2^ = 0.967) for observing the contact angles on the surfaces, and two other sets of surfaces for finer and larger scales: 130 µm^2^ scale (R^2^ = 0.62) and the 13,000 µm^2^ scale (R^2^ = 0.73). Surfaces at the smallest scale can have some microgeometric features that do not correlate with wetting while leading to a slight smoothing of the surface at a larger scale; their elimination or smoothing allows for a better observation of wetting phenomena. Only those microgeometric features that affect wetting are then taken into account. At larger scales, the surfaces are visibly too geometrically homogenized, and the surface texture is disturbed, which becomes similar for all analyzed surfaces. Then, it is difficult to distinguish contact angles on such similar geometrical surfaces. Finding the best scales for observing contact angles on surfaces is therefore an important step in wetting studies.

## 4. Conclusions

This study addresses the multiscale (area-scale and length-scale) analyses for wetting irregular micro-textured surfaces. Based on the research results and discussions, it is possible to conclude that the microtexturing of surfaces via electroerosion allows us to determine the topographic characterization parameters of the surface (the R^2^ between the discharge energy and surface topography parameters Sa, Sq, Sv, and Sz is greater than 0.8). In addition, multiscale parameters (relative area, relative length, area-scale fractal complexity, and length-scale complexity) allow for the discrimination of micro-textured EDM surfaces at specific scales.

The microtexturing of surfaces allows for modeling the wettability, including influencing the hydrophilic (68°)–hydrophobic transition (137°). Microtextured surfaces can be characterized by conventional surface characterization and multiscale parameters. Convectional surface characterization parameters that have the greatest influence on wetting are Sa, Sq, Sp, Sz, Sdq, and Sdr (R^2^ > 0.8 between contact angles and surface topographic characterization parameters). Static contact angle measurements are greatly influenced by the time from drop deposition on the surface and less by the droplet volume (1, 2, 3 µL). Together with contact angle measurements, surface free energy is often also calculated, which, in addition to surface microgeometry, also influences wettability. Surface free energy decreases with the increase in the size of craters created by electrical discharges.

Area-scale and length-scale analyses are effective in determining the best scales for observing wetting phenomena on microtextured surfaces. The best scale for observing contact angles on microtextured surfaces, according to area-scale analysis, is 1024 µm^2^ (the R^2^ between contact angles and area-scale fractal complexities is 0.967), while for the length-scale analysis, it is 32 µm (Between contact angles and length-scale fractal complexity, the largest R^2^ is 0.97 at this scale). The contact angle increases with the increasing fractal complexity of the surface. In addition, too-fine microgeometric features of the surface at fine scales do not correlate with wetting, nor do significant smoothing and simplification of the surface at large scales. There is a range of scales between fine and large that constitutes a middle compromise.

## Figures and Tables

**Figure 1 materials-17-05716-f001:**
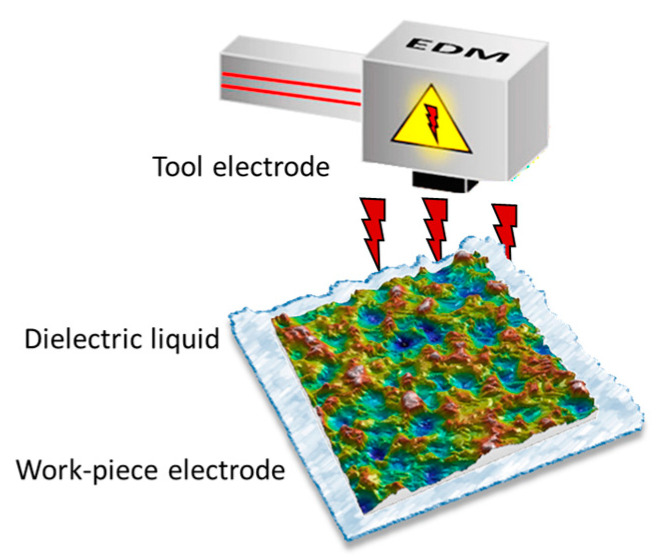
The schematic view of the EDM.

**Figure 2 materials-17-05716-f002:**
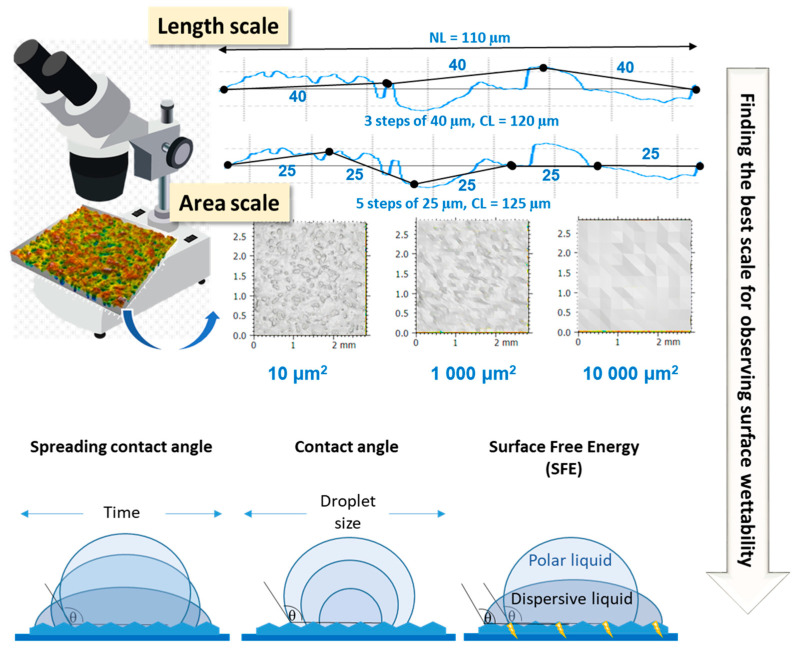
Graphical representation of experiments in these studies.

**Figure 3 materials-17-05716-f003:**
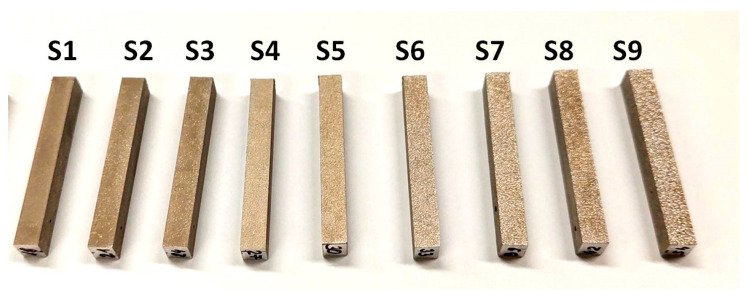
Electrical discharge machined parts.

**Figure 4 materials-17-05716-f004:**
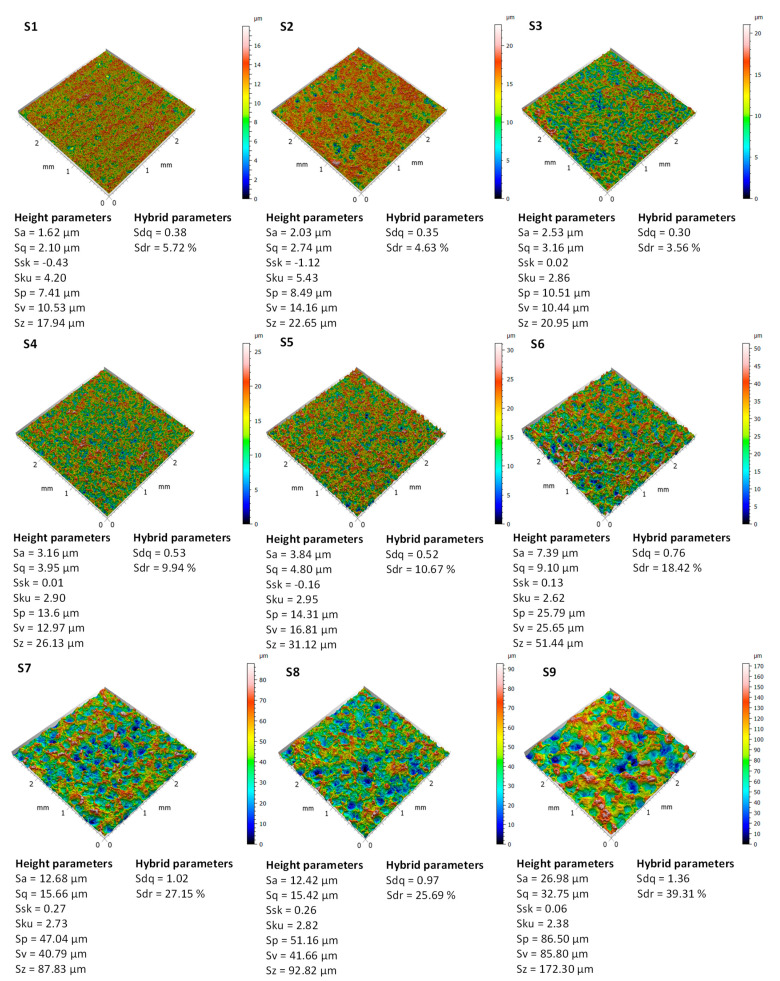
Three-dimensional images for S1–S9 EDMed surfaces.

**Figure 5 materials-17-05716-f005:**
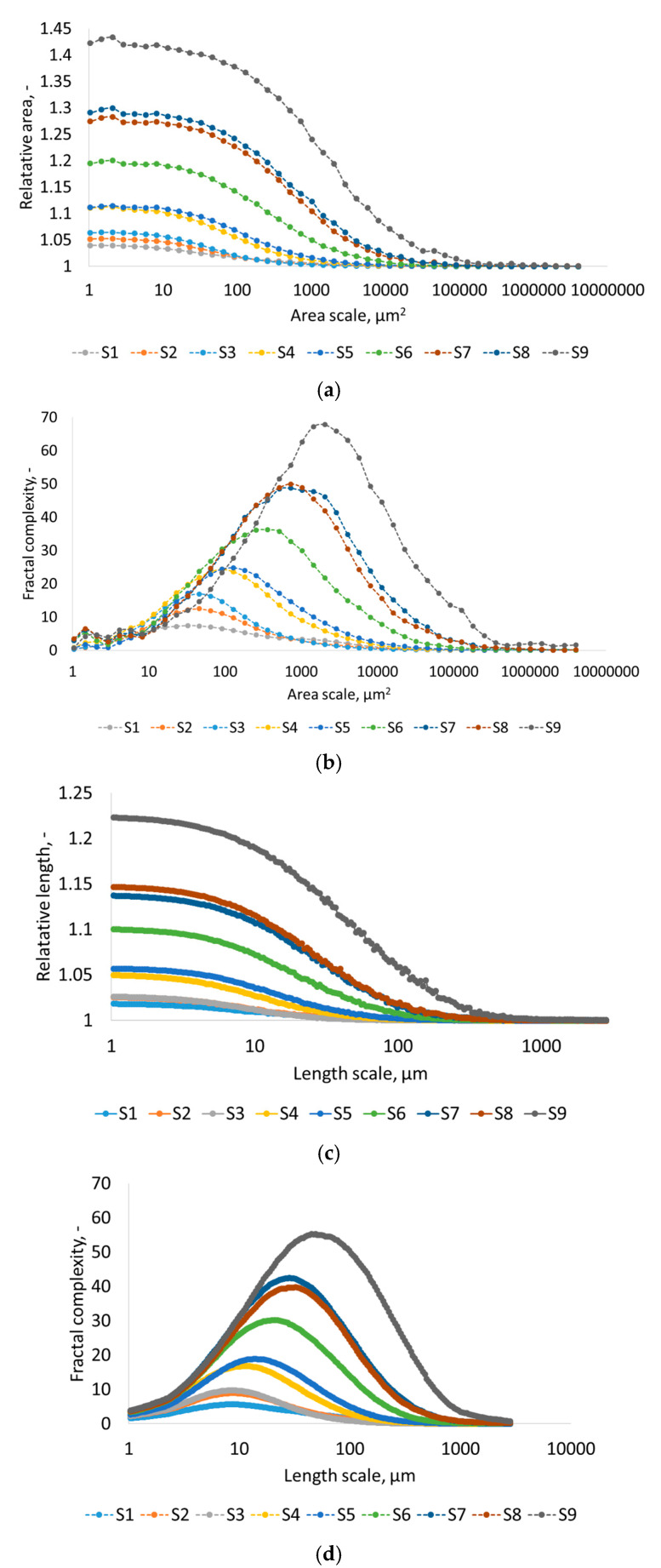
The scale-sensitive parameters: (**a**) relative area vs. scale, (**b**) area-scale fractal complexity vs. scale, (**c**) relative length vs. scale, and (**d**) length-scale fractal complexity vs. scale.

**Figure 6 materials-17-05716-f006:**
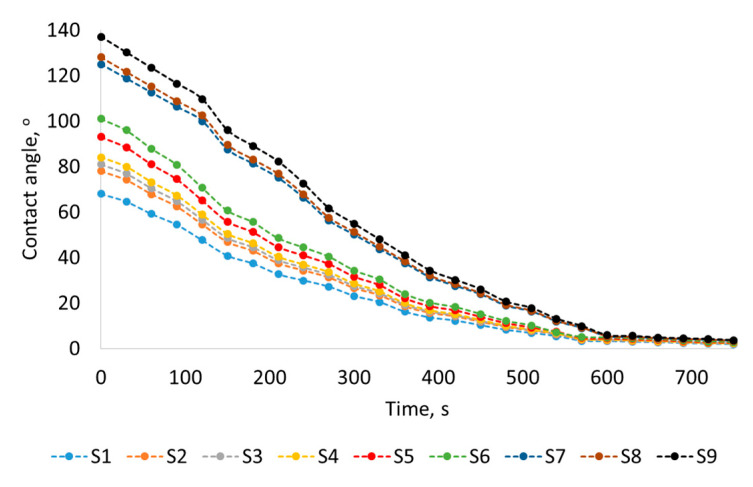
Spreading contact angle versus time.

**Figure 7 materials-17-05716-f007:**
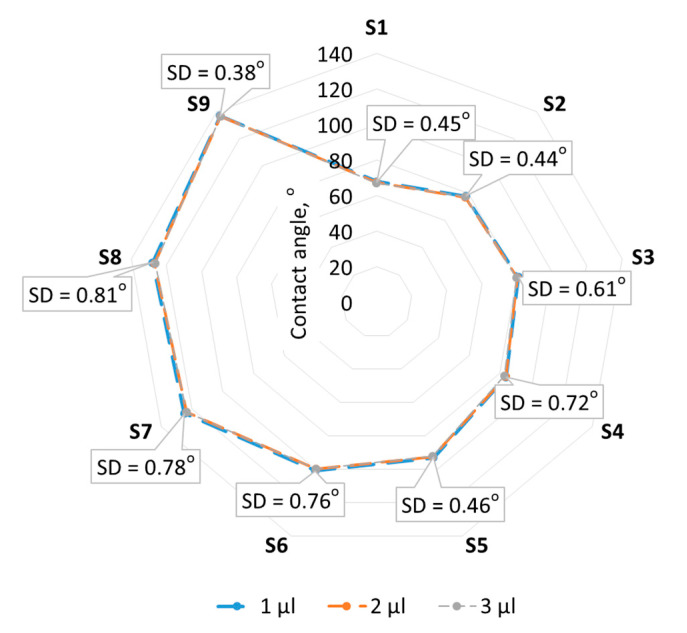
Contact angle versus droplet volume.

**Figure 8 materials-17-05716-f008:**
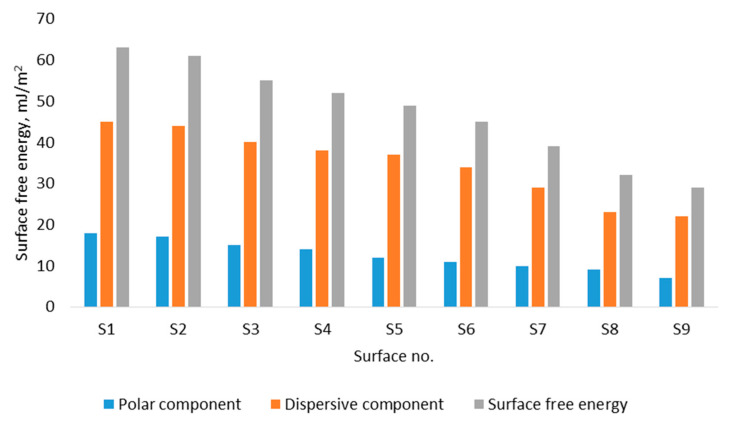
Surface free energy of electrical discharge machined surfaces.

**Figure 9 materials-17-05716-f009:**
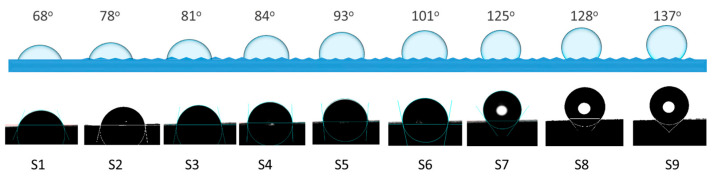
Contact angles of surfaces after EDM: outlined droplet shapes with Young-Laplace fit (**Top**); real images (**Bottom**).

**Figure 10 materials-17-05716-f010:**
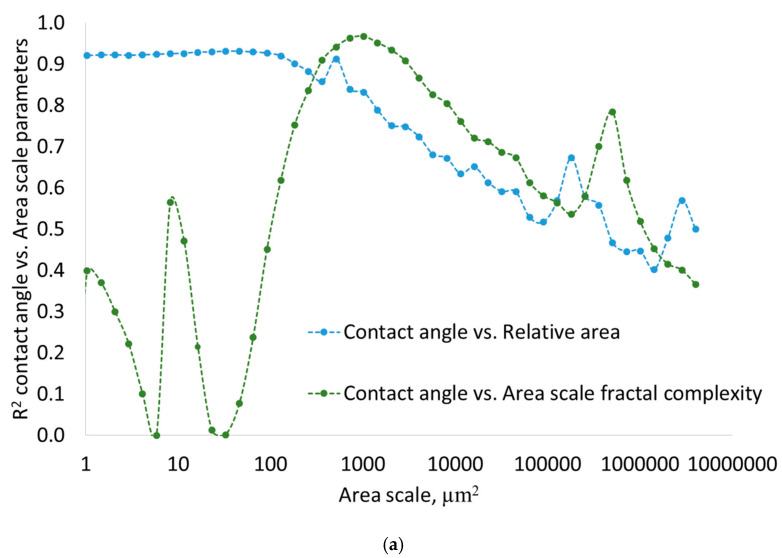
Coefficient of determination R^2^ between wetting properties and multiscale surface parameters: (**a**) area-scale, (**b**) length-scale.

**Figure 11 materials-17-05716-f011:**
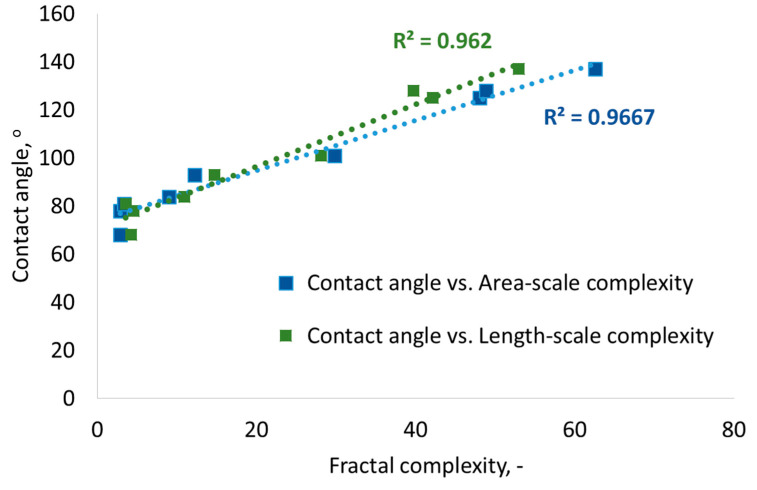
Linear trend of the relationship between contact angle and surface fractal complexity for the best observation scales.

**Figure 12 materials-17-05716-f012:**
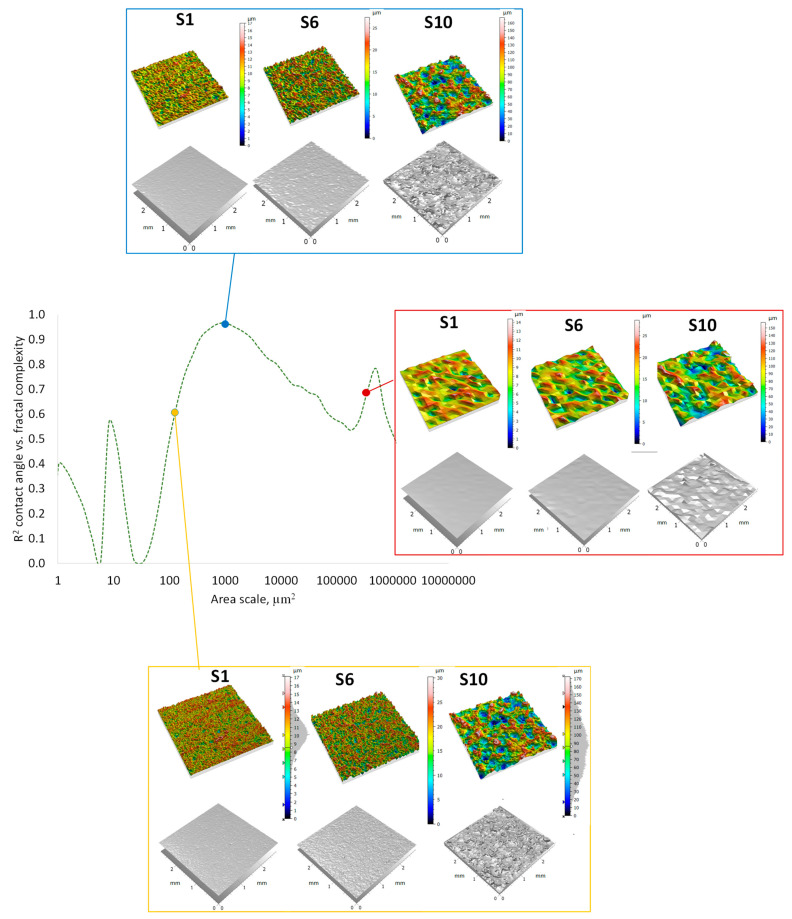
View of 3D images of surfaces (S1, S6, and S10) and their corresponding triangle meshes for different wetting observation scales.

**Table 1 materials-17-05716-t001:** EDM parameters for surface texturing and calculated discharge energy per single pulse.

Surface No.	EDM Parameters						Discharge Energy Per Single Pulse, µJ
	Pulse Peak Current, A	Pulse Peak Voltage, V	Pulse on Time, µs	Pulse Off Time, µs	Gap Size, mm	Polarity	
S1	2.8	180	7.5	15.4	0.06	+	3780
S2	4	180	10	17.8	0.07	+	7200
S3	5.5	180	11.5	20.5	0.09	+	11,385
S4	10	100	17.8	31.6	0.15	+	17,800
S5	13	100	23.7	36.5	0.18	+	30,810
S6	17	100	48.7	37	0.22	+	82,790
S7	29	100	75	42	0.34	+	217,500
S8	39	100	133.4	48.7	0.45	+	520,260
S9	72	100	487	100	0.88	+	3,506,400

**Table 2 materials-17-05716-t002:** Coefficients of determination R^2^ between discharge energy per single pulse E and surface topographic characterization parameters. Strong dependence between determinants occurs for R^2^ > 0.8 (green), medium between 0.5 and 0.8 (yellow), and low below 0.5 (red).

	S1	S2	S3	S4	S5	S6	S7	S8	S9	R^2^
Sa, µm	1.62	2.03	2.53	3.16	3.84	7.39	12.68	12.42	26.98	0.83
Sq, µm	2.1	2.74	3.16	3.95	4.8	9.10	15.66	15.42	32.75	0.83
Ssk, −	−0.43	−0.12	0.02	0.01	−0.16	0.13	0.27	0.26	0.06	0.04
Sku, −	4.2	5.43	2.86	2.9	2.95	2.62	2.73	2.82	2.38	0.13
Sp, µm	7.41	8.49	10.51	13.6	14.31	25.79	47.04	51.16	86.5	0.75
Sv, µm	10.53	14.16	10.44	12.97	16.81	25.65	40.79	41.66	85.8	0.85
Sz, µm	17.94	22.65	20.95	26.13	31.12	51.44	87.83	92.82	172.3	0.80
Sdq, −	0.38	0.35	0.3	0.53	0.52	0.76	1.02	0.97	1.36	0.59
Sdr, %	5.72	4.63	3.56	9.94	10.67	18.42	27.15	25.69	39.31	0.61
E, µJ	3780	7200	11,385	17,800	30,810	82,790	217,500	520,260	3,506,400	

**Table 3 materials-17-05716-t003:** Coefficients of determination R^2^ between contact angles (CAs) and surface topographic characterization parameters. Strong dependence between determinants occurs for R^2^ > 0.8 (green), medium for R^2^ between 0.5 and 0.8 (yellow), and low for R^2^ below 0.5 (red).

	S1	S2	S3	S4	S5	S6	S7	S8	S9	R^2^
Sa, µm	1.62	2.03	2.53	3.16	3.84	7.39	12.68	12.42	26.98	0.82
Sq, µm	2.1	2.74	3.16	3.95	4.8	9.10	15.66	15.42	32.75	0.82
Ssk, −	−0.43	−0.12	0.02	0.01	−0.16	0.13	0.27	0.26	0.06	0.58
Sku, −	4.2	5.43	2.86	2.9	2.95	2.62	2.73	2.82	2.38	0.39
Sp, µm	7.41	8.49	10.51	13.6	14.31	25.79	47.04	51.16	86.5	0.88
Sv, µm	10.53	14.16	10.44	12.97	16.81	25.65	40.79	41.66	85.8	0.79
Sz, µm	17.94	22.65	20.95	26.13	31.12	51.44	87.83	92.82	172.3	0.84
Sdq, −	0.38	0.35	0.3	0.53	0.52	0.76	1.02	0.97	1.36	0.92
Sdr, %	5.72	4.63	3.56	9.94	10.67	18.42	27.15	25.69	39.31	0.92
CA, °	68	78	81	84	93	101	125	128	137	

## Data Availability

The original contributions presented in the study are included in the article, further inquiries can be directed to the corresponding author.

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
