# Peer review of "Multiscale Wettability of Microtextured Irregular Surfaces"

_materials, 2024, doi:10.3390/ma17235716_

Round 1
Reviewer 1 Report
Comments and Suggestions for Authors
The author studied the wetting effect on
microtextured irregular surfaces.
The contact angle with respect to the
surface complexity is analyzed.
I would recommend the manuscript for a publication
if the following comments are addressed.
It seems that the surface free energy
in Figure 7 is calculated based on the
Young's law according to the measured contact angle.
However, this kind of calculation is from
"results'' to "origin''.
I would suggest the author
to discuss another aspect for
the variation of the surface free energy,
as proposed in Phys. Rev. Lett. 132, 126202 (2024),
which should be discussed in the revision.
In Figure 5, the contact angle starts from different values at the time 0. I suppose that all the contact angle should start from 180o.
It seems that the contact angle in all cases in Figure 5 converges
to a common value close to 0o.
However, Figure 8 demonstrates another value of the contact angle
much greater than 0o.
What are the differences for the contact angle in Figure 5 and Figure 8?
In the introduction, recent reviews
about wetting phenomena (Current Opinion in Colloid & Interface Science, 59, 101574, 2022; Advanced Materials, 35, 2210745, 2023), especially, the metastable states, should be discussed.
Author Response
I greatly appreciate the constructive comments. I also thank you for the effort and time put into the thorough review of the manuscript. Each comment has been carefully considered point by point and responded. All the changes in the revised paper were colored with blue.
Comments 1: [The author studied the wetting effect on
microtextured irregular surfaces.
The contact angle with respect to the
surface complexity is analyzed.
I would recommend the manuscript for a publication
if the following comments are addressed.]
Response 1: [Thank you for recommending the manuscript for publication. I have addressed your comments.]
Comments 2: "It seems that the surface free energy
in Figure 7 is calculated based on the
Young's law according to the measured contact angle.
However, this kind of calculation is from
"results'' to "origin''.
I would suggest the author
to discuss another aspect for
the variation of the surface free energy,
as proposed in Phys. Rev. Lett. 132, 126202 (2024),
which should be discussed in the revision.]
Response 2: [Thank you for your comment. Surface free energy was calculated based on the commonly used Owens, Wendt, Rabeal and Kaelble (OWRK) method. It is known that surfaces are not perfectly smooth and texture at the microscale can affect the contact angle and related phenomena with wetting. In the revised version of the manuscript, I have added a description of the aspect affecting surface free energy, referring to the suggested citation: Phys. Rev. Lett. 132, 126202 (2024). I have also cited this paper in the revised version of the manuscript. One of the paragraphs describes the wetting model according to the classical Young's law, as well as models taking into account the microtexture of surfaces, such as Wenzel and Cassie-Baxter. To these descriptions I have added the approach presented in the suggested publication. I have added the following sentences to the manuscript:
“Another approach that takes into account surface microgeometry in wetting analysis is the surface composition concept that includes the density asymmetry of the liquid-gas phases, as well as the van der Waals interactions of the liquid-solid phases. One of the assumptions of this concept is the occurrence of multiple energy minima in the total surface free energy leading to contact angle hysteresis. This allows for the development of classical Young’s law and consideration of the density ratio, van der Waals interaction, and intermolecular potential“.]
Comments 3: [In Figure 5, the contact angle starts from different values at the time 0. I suppose that all the contact angle should start from 180o.
It seems that the contact angle in all cases in Figure 5 converges
to a common value close to 0o.
However, Figure 8 demonstrates another value of the contact angle
much greater than 0o.
What are the differences for the contact angle in Figure 5 and Figure 8?]
Response 3: [Thank you for your comment and my apologies for not being fully clear about the assumptions presented in these two plots. Figure 5 shows the change in contact angle over time for the 9 analyzed surfaces. Each surface is characterized by a different surface microgeometry, which means that the initial contact angle for different surfaces is also different. Theoretically, it could be assumed that the initial contact angle value is 180o, but the measurement was taken after the drop stabilized on the surface, so as not to take into account the dynamics of the drop falling on the surface, but its stabilized shape. Therefore, time 0 actually means the moment when the drop is already deposited on the surface, in a stable state. Therefore, the contact angle values ​​presented in Figures 5 and 8 are consistent with each other. Figure 8 shows the contact angles right after the drop stabilized on the surface from the moment it was deposited on the surface.
Referring to the publication by Butt et al., which you also suggested in this review, contact angles should be measured after reaching a stable shape, as was done in this publication.
In the revised version of the manuscript I have added an explanation:
“The time taken into account in the analysis of the spreading contact angle covers the range from the moment of stabilization of the droplet deposited on the surface (0 seconds) to 700 seconds. This range allows for the observation of the behavior of the liquid droplet from deposition to complete spreading on the surface“.]
Comments 4: [In the introduction, recent reviews
about wetting phenomena (Current Opinion in Colloid & Interface Science, 59, 101574, 2022; Advanced Materials, 35, 2210745, 2023), especially, the metastable states, should be discussed.]
Response 4: [Thank you for suggesting two publications. I have cited those publications and added the discussion in the section on metastable states.
I have added the following sentences
“One way to achieve global equilibrium of drops that tend to occur in metastable states is to subject the surface to vibrations that allow the drop to achieve a stable shape {Formatting Citation}.” - Current Opinion in Colloid & Interface Science, 59, 101574, 2022.
“Wang et al. concluded that there can be other metastable states beyond the transition from the Cassie-Baxter model to the Wenzel model caused by energy barriers depending on the droplet volume, shape, dimensions and distribution of surface heterogeneity [21]”. - Advanced Materials, 35, 2210745, 2023.]
Reviewer 2 Report
Comments and Suggestions for Authors
The author shows a set of surfaces prepared by EDM. These surfaces are comprising different surface topographies and are investigated thoroughly. Contact angle measurements are done and the results related to the surface topography in detail.
I appreciate the work and effort of the author and it was very interesting to read this well prepared manuscript.
I recommend to accept it and have only some minor input and ideas to improve it further.
Chapter 3.2.: You show the spreading of contacts angle over time, which I highly appreciate, as a lot of authors ignore this. You show that time has a huge influence. But you did not state what is the consequence for all further investigations. Hence, for all investigations shown starting from line 252, you did not state at which time after deposition of the droplet the data was taken. Please add this information.
Figure 7: The way the polar and dispersive component of the surface free energy is shown is a little misleading. It looks like there is a gradient in the two components between the surface numbers. (sample numbers). But there should be one certain value of polar as well as dispersive component per sample. I suggest changing the appearance of the figure and just plot the data for each sample and do not interconnect the data from one sample to the other with a linear fit.
Figure 11: There is a different version of the height scale used for the samples S1 and S10 for the yellow labeled data pint. I like this a lot, as it gives more information regarding the surface. So, please make the appearance equal for all the height scales and I would assume to use the one giving more details. Perhaps you could use them in figure 3, too?
Both, British and American English writing of aluminium / aluminum can be found in this article. Please check and correct this.
Author Response
I greatly appreciate the constructive comments. I also thank you for the effort and time put into the thorough review of the manuscript. Each comment has been carefully considered point by point and responded. All the changes in the revised paper were colored with blue.
Comments 1: [The author shows a set of surfaces prepared by EDM. These surfaces are comprising different surface topographies and are investigated thoroughly. Contact angle measurements are done and the results related to the surface topography in detail.
I appreciate the work and effort of the author and it was very interesting to read this well prepared manuscript.
I recommend to accept it and have only some minor input and ideas to improve it further.]
Response 1: [Thank you for your positive feedback on this manuscript. I have added additional clarifications in the revised manuscript based on your suggestions.]
Comments 2: [Chapter 3.2.: You show the spreading of contacts angle over time, which I highly appreciate, as a lot of authors ignore this. You show that time has a huge influence. But you did not state what is the consequence for all further investigations. Hence, for all investigations shown starting from line 252, you did not state at which time after deposition of the droplet the data was taken. Please add this information.]
Response 2: [Thank you for this comment. I have added the following sentences to the manuscript:
"Contact angles are measured after stabilization of the droplet deposited on the surface. Based on the experiment, it is determined that after 2 seconds from the moment of contact of the droplet with the surface it reaches a stable shape."]
Comments 3: [Figure 7: The way the polar and dispersive component of the surface free energy is shown is a little misleading. It looks like there is a gradient in the two components between the surface numbers. (sample numbers). But there should be one certain value of polar as well as dispersive component per sample. I suggest changing the appearance of the figure and just plot the data for each sample and do not interconnect the data from one sample to the other with a linear fit.]
Response 3: [Thank you for this suggestion. I have corrected the Figure 7.]
Comments 4: [Figure 11: There is a different version of the height scale used for the samples S1 and S10 for the yellow labeled data pint. I like this a lot, as it gives more information regarding the surface. So, please make the appearance equal for all the height scales and I would assume to use the one giving more details. Perhaps you could use them in figure 3, too?]
Response 4: [Thank you for your comment. Figure 3 shows the surface topography at the smallest possible scale so that all microgeometric features of the surface are as visible as possible. Figure 11 shows three sample surfaces at different scales to see the effect of scale on wettability.]
Comments 5: [Both, British and American English writing of aluminium / aluminum can be found in this article. Please check and correct this.]
Response 5: [Thank you for noticing my oversight. I have corrected it in the manuscript.]
Reviewer 3 Report
Comments and Suggestions for Authors
The manuscript analyzed the contact angles correlated with surface complexity of AA 6060 after electro-discharge machining at different observation scales. The research focus is on the methodology of selecting the best scales for observing wetting phenomena on irregular surfaces, as well as indicating the topographic characterization parameters of the surface in relation to the scales. The manuscript has some research value, while some revisions are needed:
1. The Introduction section is too long. Please focus on providing literature review for the existing research work related to the author’s work. The basic introductions for contact angle, surface energy, texturing should be significantly shortened.
2. The novelty of the work should be highlighted. It is hard to find what the novelty is in the current manuscript.
3. A process schematic should be provided for the EDM process.
4. Scale bar should be added in the microscopic images in Figure 1.
5. Why did the different EDM parameters result in different surface wettabilities? Please add explanations in the revised manuscript.
6. In Figure 11, the area-scale should be different for the three figures. Why do the scales on the X axis and Y axis look the same?
7. The Discussion section is hard to follow? The authors should consider merging the Discussion section with the Results section.
8. The Conclusions section should be rewritten. Merge them into three paragraphs.
Author Response
I greatly appreciate the constructive comments. I also thank you for the effort and time put into the thorough review of the manuscript. Each comment has been carefully considered point by point and responded. All the changes in the revised paper were colored with blue.
The manuscript analyzed the contact angles correlated with surface complexity of AA 6060 after electro-discharge machining at different observation scales. The research focus is on the methodology of selecting the best scales for observing wetting phenomena on irregular surfaces, as well as indicating the topographic characterization parameters of the surface in relation to the scales. The manuscript has some research value, while some revisions are needed:
Comments 1:
- The Introduction section is too long. Please focus on providing literature review for the existing research work related to the author’s work. The basic introductions for contact angle, surface energy, texturing should be significantly shortened.
Response 1:
Thank you for this comment. Of course I find it valuable to present the most important content in the most concise way possible. However, considering all the reviews that asked me to add an even larger literature review, I am working on finding a balance between these requirements. Thank you for your understanding.
Comments 2:
- The novelty of the work should be highlighted. It is hard to find what the novelty is in the current manuscript.
Response 2:
Thank you for this comment. I have added sentences regarding novelty of the research.
“The novelty of the research includes the identification of the best scales for observing the wetting phenomena of irregular textured surfaces.”
Comments 3:
- A process schematic should be provided for the EDM process.
Response 3:
Thank you for the suggestion. I have added the EDM schematic view to the manuscript.
Comments 4:
- Scale bar should be added in the microscopic images in Figure 1.
Response 4:
This figure shows a generalized scheme of the research plan rather than the results being analyzed. Detailed descriptions of exactly these surfaces and others are presented in Figure 4, where scale bars are presented.
Comments 5:
- Why did the different EDM parameters result in different surface wettabilities? Please add explanations in the revised manuscript.
Response 5:
Thanks for your comment. The paper explains that EDM parameters, and the discharge energy calculated based on them, affect the size of surface microgeometry. One of the factors influencing wettability is the surface microgeometry. Therefore, different EDM parameters, different discharge energy, modify the microgeometry of the surface in such a way that it affects the different behavior of liquid droplets on these surfaces. More detailed explanations are provided in the manuscript.
Comments 6:
- In Figure 11, the area-scale should be different for the three figures. Why do the scales on the X axis and Y axis look the same?
Response 6:
Figure 11 (now figure 12) shows several sets of surfaces at different observation scales (area-scales). When talking about observation scales, we mean different sizes of triangles that cover the surface. In this publication, I determined which observation scale and at the same time the size of the triangle mesh best correlates with surface wetting.
Comments 7:
- The Discussion section is hard to follow? The authors should consider merging the Discussion section with the Results section.
Response 7:
Thank you for your valuable comment. In this case, I tried to follow the journal template, which suggests separating the Results and Discussion sections into two separate sections. I have revised the issues discussed in the discussion according to the order of results presented in the Results section to improve the clarity of the content.
Comments 8:
- The Conclusions section should be rewritten. Merge them into three paragraphs.
Response 8:
Thank you. However, I am more convinced to leave the conclusions as they are now. I think that shorter content from dashes clarifies the message better than three, longer paragraphs. I have presented the order of dashes from most important to least important.
Round 2
Reviewer 1 Report
Comments and Suggestions for Authors The author has addressed all my previous concerns.
I would recommend it for publication in its current form.
Author Response
Dear Reviewer,
thank you for accepting this manuscript and thank you again for all your comments.
Reviewer 3 Report
Comments and Suggestions for Authors
Some key comments have not been addressed by the authors. The authors should more seriously consider taking the review comments into consideration.
Author Response
Comments 1:
Some key comments have not been addressed by the authors. The authors should more seriously consider taking the review comments into consideration.
Response 1:
Dear Reviewer,
thank you for your feedback. I have addressed all of your comments in the new version of the manuscript. Please see the more detailed responses to your comments, as well as all of these suggestions made in the manuscript.
The manuscript analyzed the contact angles correlated with surface complexity of AA 6060 after electro-discharge machining at different observation scales. The research focus is on the methodology of selecting the best scales for observing wetting phenomena on irregular surfaces, as well as indicating the topographic characterization parameters of the surface in relation to the scales. The manuscript has some research value, while some revisions are needed:
Comments 1:
- The Introduction section is too long. Please focus on providing literature review for the existing research work related to the author’s work. The basic introductions for contact angle, surface energy, texturing should be significantly shortened.
Response 1:
I have revised the Introduction section again and shortened the content on the basics of contact angle, surface energy, and texturing. The basic introductions have been significantly shortened. 433 words (25% of the content) have been removed from the Introduction section.
However, some content, related to the contact angle theory, I could not remove according to comments from another reviewer. These comments particularly concern metastable states and the wetting models of Young, Wenzel, Cassie-Baxter, and other new approaches in this field, which I was obliged to explain more.
Comments 2:
- The novelty of the work should be highlighted. It is hard to find what the novelty is in the current manuscript.
Response 2:
The novelty of the work is highlighted in the last paragraph in the introduction section. This paragraph has been rewritten to best reflect the novelty of the work. The current version of the paragraph is as follows:
“The novelty of the research includes the identification of the best scales for observing the wetting phenomena of irregular textured surfaces. This approach includes the indication of scale-sensitive dependencies between the fractal complexity of irregular surfaces and contact angle. It was also considered important to indicate the scales of surface wetting observations in linear and areal terms. The aim of the research includes the indication of determination coefficients R2 between conventional (ISO 25178-2) and multiscale parameters (ASME B46.1) of surface topographic characterization and contact angle. The coefficients of determination between the discharge energy and the parameters of the surface topographic characterization are also calculated. The scope of the determination coefficient analyses allows to indicate the interactions of three dependent components: texturing-surface-wettability. The work also includes an analysis of factors that have a significant impact on surface wettability interactions, such as droplet spreading time on the surface and droplet volume“.
Comments 3:
- A process schematic should be provided for the EDM process.
Response 3:
Figure 1 has been added, presenting the EDM process schematic view.
Comments 4:
- Scale bar should be added in the microscopic images in Figure 1.
Response 4:
Currently it is Figure 2, and it has been corrected.
Comments 5:
- Why did the different EDM parameters result in different surface wettabilities? Please add explanations in the revised manuscript.
Response 5:
The following explanation is added to the manuscript:
“EDM parameters, mainly discharge energy calculated based on them, affect the surface microgeometry. One of the factors influencing wettability is the surface microgeometry. Therefore, different EDM parameters modify the microgeometry of the surface and affects the different behavior of liquid droplets on these surfaces. Moreover, the calculated coefficients of determination between the interactions: discharge energy in the texturing process-conventional and multiscale parameters of surface characterization-wetting, indicate the existence of these interdependencies.”
There is also added more explanation in the manuscript.
Comments 6:
- In Figure 11, the area-scale should be different for the three figures. Why do the scales on the X axis and Y axis look the same?
Response 6:
Currently it is Figure 12, and this figure is correct.
The concept of scale in surface metrology can be interpreted as the ratio of lengths on rendered surface images to actual lengths on actual surfaces. This is probably the definition that the reviewer is asking about. However, there is also another meaning of scale used in scale-sensitive fractal analyses and more broadly described in standard ASME B46.1 and also by the creator of this method in publications, as an example - doi: 10.1016/j.cirp.2018.06.001. According to the assumptions of this approach, scale refers to a segment of wavelengths or spatial frequencies.
Accordingly, Figure 12 presents three identical surfaces at different scales using the area-scale method. Each surface is built of triangular tiles with a fixed area (certain scale). It turns out that depending on the way of simulating the surface with different sizes of triangles, different values ​​of surface wetting can be obtained. Interestingly, when analyzing irregular surfaces, these are not the smallest triangles covering the surface, nor the largest ones. There is a certain scale of observation, and thus the size of the triangles, which best allows for the correlation of surface complexity parameters with the contact angle.
More explanation for understanding the scale in the context of this publication has been added to the manuscript. Also the discussion of figure 12 is more extensive in the revised manuscript.
Comments 7:
- The Discussion section is hard to follow? The authors should consider merging the Discussion section with the Results section.
Response 7:
The Discussion and Results sections are merged in the revised version of the manuscript.
Comments 8:
- The Conclusions section should be rewritten. Merge them into three paragraphs.
Response 8:
The Conclusions section has been rewritten to contain 3 paragraphs.